# Lifelong Sequence Generation with Dynamic Module Expansion and Adaptation

**Chengwei Qin♣, Chen Chen♣, Shafiq Joty♣♠**
♣ Nanyang Technological University
♠ Salesforce AI
{chengwei003@e.ntu, chen1436@e.ntu, srjoty@ntu}.edu.sg

## Abstract

Lifelong sequence generation (LSG), a problem in continual learning, aims to continually train a model on a sequence of generation tasks to learn constantly emerging new generation patterns while avoiding the forgetting of previous knowledge. Existing LSG methods mainly focus on maintaining old knowledge while paying little attention to knowledge transfer across tasks. In contrast, humans can better learn new tasks by leveraging previously acquired knowledge from similar tasks. Inspired by the learning paradigm of humans, we propose Dynamic Module Expansion and Adaptation (DMEA), which enables the model to dynamically determine the architecture for acquiring new knowledge based on task correlation and select the most similar previous tasks to facilitate adaptation to new tasks. In addition, as the learning process can easily be biased towards the current task which might cause more severe forgetting of previously learned knowledge, we propose dynamic gradient scaling to balance the learning of the current task and replayed tasks. With extensive experiments, we demonstrate that DMEA can consistently outperform existing methods in different LSG settings.

## 1 Introduction

With the recent advancements in pre-trained language models (LMs), current sequence generation methods have achieved impressive performance on a variety of generation tasks (Radford et al., 2019; Raffel et al., 2020). Typically, these models are trained on a fixed corpus, assuming the underlying data distribution to be static (Ham et al., 2020; El-Kassas et al., 2021). However, real cognitive tasks are generally more complex involving changing contexts and dynamic environments. The ever-changing data distribution causes the models to face challenges in acquiring new knowledge, while retaining the prior knowledge. Speaking about what is next for NLP, Kathleen McKeown

in a recent interview said: "Most models are static. But the world changes every minute, every second. Dealing with a dynamic world is a new area that's up and coming." (Source)

A potential solution is to formalize sequence generation as lifelong sequence generation or LSG (Sun et al., 2020), where the model is expected to learn sequentially from a stream of generation tasks with potentially different data distributions. In such cases of distribution shift, the model might forget previously acquired knowledge upon learning new tasks, a phenomenon known as *catastrophic forgetting* (McCloskey and Cohen, 1989). Previous LSG methods (Mi et al., 2020; Sun et al., 2020; Madotto et al., 2021) mainly explore different ways to alleviate forgetting. Recently, Zhang et al. (2022) propose Adaptive Compositional Modules (ACM) which dynamically adds modules for new tasks depending on whether there are reusable previous modules, achieving SOTA performance on LSG.

Despite its effectiveness, ACM has several key limitations. First, it mainly focuses on mitigating forgetting of previously acquired knowledge while paying little attention to transferring learned knowledge to new tasks which is as important for continual learning as preventing forgetting (Ke et al., 2020). In fact, a hallmark of human intelligence is that humans can better learn new tasks by leveraging previously acquired knowledge from similar tasks (Lake et al., 2017). They can not only determine whether previously acquired skills are sufficient to solve a new task, but also exploit the most similar learned skills to facilitate the learning of the task; see Appendix A.1 for an illustration. Second, ACM does not consider the correlation between learned tasks and the new task when adding modules, which might hinder finding the optimal architecture (case study in Appendix A.9). Finally, the learning process in ACM can be biased towards the new task as the gradient norm of the new task on reused modules is typically much larger than

that of replayed tasks, which may affect previously acquired knowledge; see Appendix A.2 for an explanation.

Inspired by the learning paradigm of humans and to address the above limitations of ACM, in this work we propose Dynamic Module[1] Expansion and Adaptation (DMEA). We divide the learning process of a new task into three stages: expansion, selection and adaptation. In the expansion stage, DMEA determines whether to reuse modules of previous tasks or insert new modules for learning novel knowledge. Inspired by Zhang et al. (2022), it utilizes differentiable architecture search (Liu et al., 2019) to enable the model to dynamically determine the architecture for solving the new task. The learnable coefficients in architecture search are initialized based on the cosine similarity of word frequency distributions between learned tasks and the new task, aiming to discover the optimal model architecture. After searching, the module with the largest coefficient in every layer is chosen for the new task. In the selection stage, DMEA selects the top-$K$ most similar previous tasks through input subspace (Lin et al., 2022b). Finally, in the adaptation stage, it utilizes the selected similar tasks to facilitate adaptation to the new task. The output of selected similar tasks is fused with that of the new task using learnable coefficients in every transformer layer to enable forward knowledge transfer. This is indeed an instance of mixture-of-experts (Masoudnia and Ebrahimpour, 2014).

In addition, when the model learns a new task, DMEA also incorporates pseudo-sample replay (Sun et al., 2020) to further mitigate catastrophic forgetting. To address the "bias to the new task" in the gradient update, we introduce dynamic gradient scaling to balance the learning of the new task and replayed tasks. To verify the effectiveness of DMEA, we conduct extensive experiments on various generation tasks in different LSG settings. The empirical results show that DMEA can consistently outperform previous state-of-the-art baselines.

In summary, our main contributions are:

- To the best of our knowledge, we are the first to explore solving LSG from the perspective of human learning. We propose DMEA, a novel method based on dynamic module expansion and adaptation, to alleviate catastrophic forgetting and facilitate knowledge transfer in LSG.

- With extensive experiments and analysis, we demonstrate the effectiveness of our method compared to existing ones in different LSG settings.

## 2 Related Work

**Lifelong Learning** (LL) aims to continually learn knowledge from a sequence of tasks with different distributions. The goal is twofold: alleviate *catastrophic forgetting* (McCloskey and Cohen, 1989) of learned tasks, and facilitate knowledge transfer (Lopez-Paz and Ranzato, 2017) across tasks.

Catastrophic forgetting typically means that the model forgets previously acquired knowledge after learning new tasks. Prior LL methods mainly focus on mitigating this problem and can be divided into three categories. First, *regularization-based* methods constrain the update of parameters that are important to learned tasks to retain previous knowledge (Kirkpatrick et al., 2017; Li and Hoiem, 2017; Zenke et al., 2017; Ritter et al., 2018). Second, *architecture-based* methods dynamically adjust the model architecture to acquire new information while preventing the forgetting of previously learned tasks (Rusu et al., 2016; Chen et al., 2016; Fernando et al., 2017; Madotto et al., 2021; Zhang et al., 2022). Finally, *memory-based* methods keep a number of key samples from previous tasks in memory to alleviate forgetting (Rebuffi et al., 2017; Shin et al., 2017; Chaudhry et al., 2019; Qin and Joty, 2022a). The memory data can be either real examples (Han et al., 2020) or generated by language models (Sun et al., 2020; Qin and Joty, 2022b).

More recently, researchers have considered exploring knowledge transfer in LL, *i.e.,* learning on a task can benefit from learning on another task by transferring related knowledge. This includes CTR (Ke et al., 2021) and CUBER (Lin et al., 2022a). Despite their effectiveness, these methods mainly focus on classification tasks, while generation tasks typically have more complex label space. Note that this line of research is different from transfer learning (Ruder et al., 2019), which mainly focuses on exploring better ways to reuse learned knowledge which is usually static, *e.g.,* a frozen language model. In contrast, the acquired knowledge is continually accumulated in lifelong learning.

**Lifelong Sequence Generation** (LSG) enables the model to learn sequentially from a stream of generation tasks. Sun et al. (2020) propose LAMOL which formalizes different types of tasks as ques-

---

[1]Following Zhang et al. (2022), we use an Adapter (Houlsby et al., 2019) as the insertable module.

tion answering and utilizes pseudo-sample replay to alleviate forgetting. Chuang et al. (2020) further improve LAMOL by knowledge distillation (Hinton et al., 2015). AdapterCL (Madotto et al., 2021) inserts task-specific modules into every transformer layer to learn new tasks while keeping the pre-trained LM and previous modules frozen. On the basis of AdapterCL, Zhang et al. (2022) introduce ACM which dynamically adds modules for learning new tasks depending on whether there are reusable previously inserted modules. Though ACM can enable knowledge transfer to some extent via module sharing, there is no explicit mechanism to encourage knowledge transfer across tasks, a common phenomenon of human learning.

**Summary.** Existing work in LSG mainly focuses on mitigating the catastrophic forgetting of previously learned knowledge while paying little attention to knowledge transfer across tasks. In contrast to these lines of work, we aim to explicitly encourage forward knowledge transfer in LSG inspired by the way humans learn (Lake et al., 2017).

## 3 Problem Formulation

LSG involves learning from a stream of sequence generation tasks $\mathbb{T} = (\mathcal{T}^1, ..., \mathcal{T}^n)$, where every task $\mathcal{T}^i$ has its own training set $D^i_{\text{train}}$, validation set $D^i_{\text{valid}}$, and test set $D^i_{\text{test}}$. Every dataset $D$ contains a set of examples $\{(X_j, Y_j)\}_{j=1}^{|D|}$, where $X_j$ and $Y_j$ denote the input and output texts, respectively. At time step $k$, the model is trained on the training set $D^k_{\text{train}}$ of task $\mathcal{T}^k$ and has no access to real samples of previously learned tasks.

After the training on $D^k_{\text{train}}$, the model is expected to perform well on all the tasks learned so far, *i.e.*, $\mathcal{T}^1, ..., \mathcal{T}^k$, and will be evaluated on the test set $D^i_{\text{test}}$ of each task $\mathcal{T}^i(1 \le i \le k)$ with corresponding evaluation metrics separately. Therefore, to achieve the goal of LSG, the model is required to alleviate the forgetting of acquired knowledge and better learn new patterns through possible forward knowledge transfer.

### 3.1 Data Format

Given an input-output text pair $(X, Y)$ for a task, the model learns to decode the output text $Y$ after reading the input $X$. Following Zhang et al. (2022), a natural language question $Q$ describing the purpose of each task (task instruction) is inserted after the input to form a triple $(X, Q, Y)$; see Appendix A.3 for an example. To learn a new

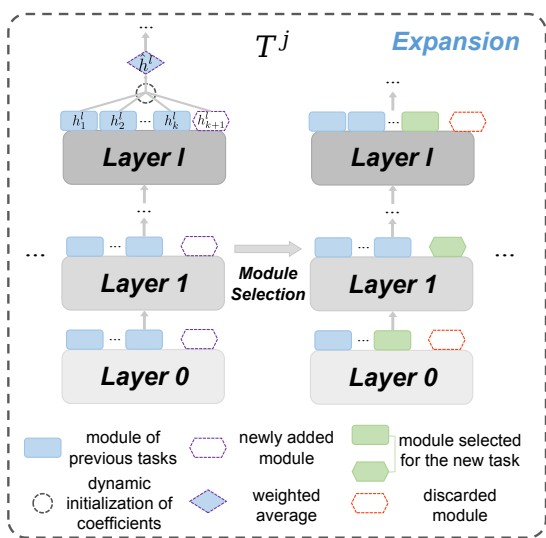

Figure 1: In the **expansion** stage, after inserting a new module (purple dashed hexagon) into each layer, DMEA dynamically determines the architecture by differentiable architecture search. Specifically, the outputs of all modules in the same layer are fused through dynamically initialized learnable coefficients. The weighted average is then passed to the next layer of the model for learning. Note that only newly added modules (purple dashed hexagons) are learnable modules in this stage. After several epochs of training, the module with the *largest* coefficient in every layer (green polygon) is selected for the new task. The selected module can be either a previous module (rectangle) or the newly added one (hexagon). Finally, newly added modules that are not selected (red dashed hexagons) will be discarded.

task, the model is optimized to decode $Y$ given $X$ and $Q$. Denoting the concatenation of $X, Q$ and $Y$ as $A$, the autoregressive training objective is:

$$\mathcal{L}_{\text{task}} = -\sum_{j=m+1}^{n} \log p_\theta(A_j|A_{<j}) \qquad (1)$$

where $n$ is the total number of tokens in $A$ and $(A_1, ..., A_m)$ is the concatenation of $X$ and $Q$, and $\theta$ denotes the model parameters.

## 4 Methodology

Inspired by how humans learn a new task (Fig. 5), DMEA divides the learning process into three stages. The *expansion* stage (§4.1) first determines the model architecture dynamically. The *selection* stage (§4.2) then selects the top-K most similar previous tasks which are utilized in the final *adaptation* stage (§4.3) to facilitate adaptation to the new task. We also employ pseudo-sample replay along with a dynamic gradient scaling method to balance the learning of the new and replayed tasks.

## 4.1 Expansion Stage

Humans are able to determine whether previously acquired skills are sufficient to solve a new task. Our method DMEA aims to mimic this learning process in the expansion stage. It can dynamically decide whether to reuse modules of previous tasks or insert a new module in every transformer layer to learn novel knowledge. Inspired by Zhang et al. (2022), we utilize differentiable architecture search (Liu et al., 2019) to achieve this goal.

Specifically, assuming that there are $k$ modules (*i.e.,* Adapter (Houlsby et al., 2019)) $\{m_1^l, ..., m_k^l\}$ in layer $l$ of the transformer model before learning a new task $\mathcal{T}^j$, we temporarily insert a new module $m_{k+1}^l$ into this layer at the beginning of the expansion stage. For each forward pass, after calculating the output $h_t^l$ of every module $m_t^l$ in the layer separately, we fuse all outputs $\{h_1^l, ..., h_{k+1}^l\}$ through learnable coefficients $\{\lambda_1^l, ..., \lambda_{k+1}^l\}$ as follows.

$$\hat{h}^l = \sum_{t=1}^{k+1} \frac{e^{\lambda_t^l}}{\sum_{s=1}^{k+1} e^{\lambda_s^l}} h_t^l \qquad (2)$$

The weighted average $\hat{h}^l$ is then passed to the next part of the model for learning. After training the model on $D_{\text{train}}^j$ for several epochs using $\mathcal{L}_{\text{train}}$ (defined in §4.3), we select the module with the **largest** coefficient in every layer for the new task $\mathcal{T}^j$.

Different from Zhang et al. (2022) which initialize $\{\lambda_1^l, ..., \lambda_{k+1}^l\}$ with predefined hyperparameters, we propose to dynamically initialize learnable coefficients based on the correlation between the learned tasks $\mathcal{T}^1, ..., \mathcal{T}^{j-1}$ and new task $\mathcal{T}^j$. Denoting the word frequency distribution of $\mathcal{T}^i$ as $f^i$ and all previous tasks sharing the module $m_t^l$ as $\mathcal{Z}_t^l$, the learnable coefficient $\lambda_t^l$ is initialized as:

$$\lambda_t^l = \begin{cases} \max_{\mathcal{T}^i \in \mathcal{Z}_t^l} \cos(f^i, f^{k+1}), & 1 \le t \le k \\ \min_{1 \le i \le k} \lambda_i^l, & t = k+1 \end{cases} \qquad (3)$$

where cos is the cosine similarity function and $f^i$ is calculated based on the training set $D_{\text{train}}^i$. In this way, a previous module shared by tasks with higher word frequency distribution similarity to the new task has a larger initial coefficient, increasing the tendency to reuse it. In addition, the coefficient $\lambda_{k+1}^l$ of the newly added module $m_{k+1}^l$ is initialized to the minimum value of the initial coefficients $\{\lambda_1^l, ..., \lambda_k^l\}$ of previously added modules $\{m_1^l, ..., m_k^l\}$ to encourage module reuse.

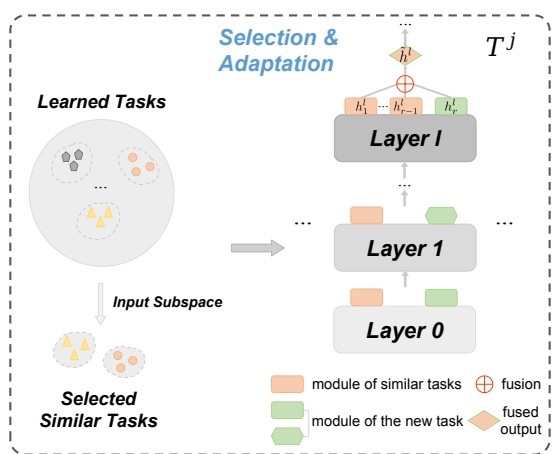

Figure 2: In the **selection** stage, DMEA selects the top-$K$ most similar previous tasks through input subspace to facilitate adaptation to a new task. During **adaptation**, the output of the selected similar tasks is fused with that of the new task in every layer to enable forward knowledge transfer. Note that only modules selected for the new task (green polygons) are learnable modules in the adaptation stage. In addition, DMEA introduces dynamic gradient scaling to balance the learning of the new task and replayed tasks.

The selected module in layer $l$ can be either from previous modules $\{m_1^l, ..., m_k^l\}$ or the newly added one $m_{k+1}^l$ and will be tuned in the adaptation stage to accommodate new knowledge. We then discard newly added modules that are not selected. Note that only newly added modules and coefficients are learnable in the expansion stage; the pre-trained LM and previous modules are kept frozen.

## 4.2 Selection Stage

As humans, we can better acquire new knowledge by recognizing and utilizing knowledge from previously learned tasks that are similar (Lake et al., 2017). Based on the observation that the norm of one task's gradient projection onto the subspace of another task can characterize the correlation between them when the model architecture is static (Lin et al., 2022b), we further extend it to dynamic modules. Specifically, we obtain the input subspace of each task using modules of it and select the top-$K$ most similar previous tasks by input subspace similarity to facilitate adaptation to the new task $\mathcal{T}^j$. The model architecture induced from the expansion stage is used for selection and adaptation.

Similar to Lin et al. (2022b), we adopt Singular Value Decomposition (SVD) to obtain the input subspace of each task. After training the model on $D_{\text{train}}^j$ for several epochs in the expansion stage, we randomly select $n$ samples

$\{X_1, ..., X_n\}$ from $D_{\text{train}}^j$ and obtain their representations $\{\boldsymbol{X}_1, ..., \boldsymbol{X}_n\} \in \mathbb{R}^m$ by forward-propagating them through the network. We use the final-layer representation of the last non-padding token in the input as the sample representation.

After obtaining the representation matrix $\boldsymbol{R}^j = [\boldsymbol{X}_1, ..., \boldsymbol{X}_n] \in \mathbb{R}^{m \times n}$ for task $\mathcal{T}^j$, we apply SVD to $\boldsymbol{R}^j$, *i.e.*, $\boldsymbol{R}^j = \boldsymbol{U}^j \boldsymbol{\Sigma}^j (\boldsymbol{V}^j)'$, where $\boldsymbol{U}^j = [\boldsymbol{u}_1^j, ..., \boldsymbol{u}_m^j] \in \mathbb{R}^{m \times m}$ is composed of left-singular vectors $\boldsymbol{u}_i^j$, $\boldsymbol{\Sigma}^j \in \mathbb{R}^{m \times n}$ is a rectangular diagonal matrix with singular values on the diagonal, and $\boldsymbol{V}^j = [\boldsymbol{v}_1^j, ..., \boldsymbol{v}_n^j] \in \mathbb{R}^{n \times n}$ is composed of right-singular vectors $\boldsymbol{v}_i^j$. To obtain the input subspace $\boldsymbol{S}^j$ of $\mathcal{T}^j$, we select the first $k$ left-singular vectors in $\boldsymbol{U}^j$ to form the bases $\boldsymbol{B}^j = [\boldsymbol{u}_1^j, ..., \boldsymbol{u}_k^j]$ for $\boldsymbol{S}^j$, where $k$ is determined by the requirement: $||\boldsymbol{R}_k^j||_F^2 \geq \epsilon^j ||\boldsymbol{R}^j||_F^2$ with $\boldsymbol{R}_k^j$ being the $k$-rank approximation of $\boldsymbol{R}^j$, $F$ being the Frobenius norm, and $\epsilon^j$ being a predefined threshold.

For the new task $\mathcal{T}^j$, the norm of its subspace projection onto the subspace of a previously learned task $\mathcal{T}^i$ could characterize the similarity $\boldsymbol{Q}_{j,i}$ between these two tasks. More formally,

$$\boldsymbol{Q}_{j,i} = \frac{||\text{Proj}_{\boldsymbol{S}^i}(\boldsymbol{S}^j)||_2}{||\boldsymbol{B}^j||_2} \qquad (4)$$

where $\text{Proj}_{\boldsymbol{S}^i}(\boldsymbol{S}^j) = \boldsymbol{B}^j \boldsymbol{B}^i (\boldsymbol{B}^i)'$ denotes the subspace projection. After getting the similarity scores $\boldsymbol{Q}_{j,i}, 1 \leq i < j$ of all previous tasks, we pick $K$ tasks $\mathbb{T}^{\text{sim}} = (\mathcal{T}^1, ..., \mathcal{T}^K)$ with the top-$K$ highest scores to facilitate adaptation to the new task $\mathcal{T}^j$.

### 4.3 Adaptation Stage

For adaptation to $\mathcal{T}^j$, assume that $\mathbb{T}^{\text{all}} = (\mathcal{T}^1, ..., \mathcal{T}^K, \mathcal{T}^j)$ contains a total of $r$ modules $\{m_1^l, ..., m_r^l\}$ in layer $l$. During the training on $D_{\text{train}}^j$ using $\mathcal{L}_{\text{train}}$ (see Eq. (7)), for each sample in $D_{\text{train}}^j$, we fuse the output $h_s^l$ of each module $m_s^l \in \{m_1^l, ..., m_r^l\}$ by learnable coefficients $\{\alpha_1^l, ..., \alpha_r^l\}$ to enable *forward knowledge transfer*:

$$\tilde{h}^l = \sum_{s=1}^{r} \frac{e^{\alpha_s^l}}{\sum_{u=1}^{r} e^{\alpha_u^l}} h_s^l \qquad (5)$$

The learnable coefficients $\{\alpha_1^l, ..., \alpha_r^l\}$ are equally initialized to 1.0. Similar to the expansion stage, the fused output $\tilde{h}^l$ is passed to the next part of the model for learning. After training, the learnable coefficients will be saved for inference. Note that we only tune modules selected in the expansion stage (can be modules of previous tasks or newly

added modules) and learnable coefficients while keeping the pre-trained language model and other modules frozen.

As there is no saved real sample of previously learned tasks when the model adapts to a new task, we also incorporate pseudo-sample replay (Sun et al., 2020) to alleviate the forgetting of acquired knowledge. We achieve this by simultaneously training the model as a task solver ($\mathcal{L}_{\text{task}}$ in §3.1) and as a data generator. When training as a data generator, the model learns to generate the triple $(X, Q, Y)$ given a task-specific generation token $G$ as input. Then before learning a new task, the model can generate pseudo samples of previous tasks, which are combined with new data for training to mitigate forgetting. Denoting the concatenation of $G, X, Q$ and $Y$ as $A'$, the data generation loss is expressed as:

$$\mathcal{L}_{\text{data}} = -\sum_{i=2}^{m} \log p_\theta(A_i' | A_{<i}') \qquad (6)$$

where $m$ is the total number of tokens in $A'$. The overall loss that DMEA optimizes for adapting to a new task is:

$$\mathcal{L}_{\text{train}} = \mathcal{L}_{\text{task}} + \mu \mathcal{L}_{\text{data}} \qquad (7)$$

where $\mu$ is the weight of data generation loss.

After the expansion stage, if the new task reuses some modules of previously learned tasks, the model will generate some pseudo samples of these tasks and train the model using $\mathcal{L}_{\text{train}}$ on the combination of new data and pseudo data. As the model has not seen new data before, the gradient norm of the new task on reused modules is much larger than that of replayed tasks. The learning process can easily be biased towards the new task which may affect previously acquired knowledge.

Therefore, to balance the learning of the new task and replayed tasks, we introduce *dynamic gradient scaling*. Specifically, assuming that the new task $\mathcal{T}^j$ reuses $s$ modules $\{m_1, ..., m_s\}$ of a previous task $\mathcal{T}^i$ in all layers, we randomly select $q$ examples from $D_{\text{train}}^j$ and pseudo samples of $\mathcal{T}^i$ separately and forwards them through the model to obtain the gradient of $\mathcal{T}^j$ and $\mathcal{T}^i$ using $\mathcal{L}_{\text{train}}$ with regard to reused modules $\{m_1, ..., m_s\}$, denoted as $g^j$ and $g^i$, respectively. The dynamic scale factor $\eta_t^i$ is then calculated as:

$$\eta_t^i = (\frac{||g^j||_2}{||g^i||_2} - 1)e^{-t} + 1 \qquad (8)$$

| Methods | | Finetune | EWC | LAMOL | Metac | Adapter +LAMOL | AdapterCL | ACM | DMEA | MTL |
|---|---|---|---|---|---|---|---|---|---|---|
| Tune Whole Model? | | ✓ | ✓ | ✓ | ✓ | ✗ | ✗ | ✗ | ✗ | ✓ |
| Pseudo-sample Replay? | | ✗ | ✗ | ✓ | ✓ | ✓ | ✗ | ✓ | ✓ | — |
| Similar Tasks | #1 | $42.9_{\pm0.2}$ | $56.6_{\pm0.1}$ | $65.5_{\pm0.3}$ | $65.2_{\pm0.4}$ | $64.5_{\pm0.2}$ | $63.4_{\pm0.3}$ | $64.8_{\pm0.3}$ | $\mathbf{65.8}_{\pm0.2}$ | $67.1_{\pm0.2}$ |
| | #2 | $51.8_{\pm0.1}$ | $61.0_{\pm0.2}$ | $65.2_{\pm0.4}$ | $65.0_{\pm0.2}$ | $64.6_{\pm0.4}$ | $63.4_{\pm0.3}$ | $64.9_{\pm0.2}$ | $\mathbf{65.6}_{\pm0.2}$ | — |
| | #3 | $45.2_{\pm0.2}$ | $57.8_{\pm0.1}$ | $\mathbf{65.7}_{\pm0.2}$ | $65.4_{\pm0.4}$ | $64.2_{\pm0.3}$ | $63.4_{\pm0.3}$ | $64.5_{\pm0.2}$ | $65.5_{\pm0.1}$ | — |
| | #4 | $31.4_{\pm0.4}$ | $46.6_{\pm0.3}$ | $65.6_{\pm0.3}$ | $65.5_{\pm0.1}$ | $65.1_{\pm0.1}$ | $63.4_{\pm0.3}$ | $65.4_{\pm0.3}$ | $\mathbf{66.2}_{\pm0.3}$ | — |
| Average | | $42.9_{\pm8.5}$ | $55.5_{\pm6.2}$ | $65.5_{\pm0.2}$ | $65.3_{\pm0.2}$ | $64.6_{\pm0.4}$ | $63.4_{\pm0.0}$ | $64.9_{\pm0.4}$ | $\mathbf{65.8}_{\pm0.3}$ | $67.1_{\pm0.0}$ |
| Random Tasks | #1 | $33.8_{\pm0.4}$ | $37.6_{\pm0.5}$ | $55.7_{\pm0.4}$ | $55.9_{\pm0.3}$ | $53.2_{\pm0.2}$ | $56.1_{\pm0.5}$ | $56.7_{\pm0.3}$ | $\mathbf{57.5}_{\pm0.3}$ | $59.7_{\pm0.2}$ |
| | #2 | $33.1_{\pm0.3}$ | $38.4_{\pm0.4}$ | $62.6_{\pm0.2}$ | $62.4_{\pm0.4}$ | $61.8_{\pm0.3}$ | $64.2_{\pm0.2}$ | $64.9_{\pm0.4}$ | $\mathbf{65.6}_{\pm0.3}$ | $67.5_{\pm0.1}$ |
| | #3 | $25.7_{\pm0.1}$ | $43.2_{\pm0.3}$ | $54.8_{\pm0.1}$ | $55.4_{\pm0.2}$ | $53.6_{\pm0.3}$ | $55.6_{\pm0.4}$ | $56.3_{\pm0.1}$ | $\mathbf{57.3}_{\pm0.2}$ | $60.4_{\pm0.3}$ |
| | #4 | $34.2_{\pm0.3}$ | $48.9_{\pm0.1}$ | $64.7_{\pm0.4}$ | $65.3_{\pm0.3}$ | $62.5_{\pm0.1}$ | $65.4_{\pm0.3}$ | $66.2_{\pm0.2}$ | $\mathbf{67.4}_{\pm0.1}$ | $69.8_{\pm0.1}$ |
| Average | | $31.7_{\pm4.0}$ | $42.0_{\pm5.2}$ | $59.5_{\pm4.9}$ | $59.8_{\pm4.9}$ | $57.8_{\pm5.1}$ | $60.3_{\pm5.2}$ | $61.0_{\pm5.3}$ | $\mathbf{62.0}_{\pm5.3}$ | $64.4_{\pm5.1}$ |

Table 1: The average performance score for each task sequence after learning all tasks. **Bold** indicates the best score. 'MTL' stands for 'multi-task learning', serving as the *upper bound* for LSG. In each scenario, DMEA is significantly better than ACM with $p$-value $< 0.05$ (paired t-test). Note that while LAMOL and Metac are not directly comparable to other adapter-based methods as their learnable parameters are orders of magnitude larger, DMEA still outperforms them in most cases. The comparison of learnable parameters and computational resources between ACM and DMEA is reported in Appendix A.8.

where $t$ is the number of completed training epochs. After dynamic gradient scaling, the total loss for jointly learning $\mathcal{T}^j$ and $\mathcal{T}^i$ is:

$$\mathcal{L}_{\text{total}} = \mathcal{L}_{\text{train}}^j + \eta_t^i \mathcal{L}_{\text{train}}^i \quad (9)$$

Note that in the early stage of training, the value of $t$ is small. $\eta_t$ is greater than 1 to balance the gradient of the new task $\mathcal{T}^j$ and the replayed task $\mathcal{T}^i$. When the model has seen enough new data in the late stage of training (no need to balance), $\eta_t$ is approximately equal to 1 as the value of $t$ is large.

## 5 Experimental Setup

In this section, we first describe investigated tasks and then introduce methods compared in our work.

### 5.1 Tasks

Four representative sequence generation tasks are investigated in our work: natural language generation, summarization, task-oriented dialogue and SQL query generation. Following Zhang et al. (2022), we consider two different scenarios: (*i*) LSG on *similar* tasks where the model learns a sequence of tasks of the same type but different domains, and (*ii*) LSG on *random* tasks where the model learns knowledge from different types of tasks. For LSG on similar tasks, we use five different domains from two natural language generation datasets (RNNLG (Wen et al., 2015) and E2ENLG (Novikova et al., 2017)) to form the task sequences. We further incorporate summarization (CNNDM (See et al., 2017)), task-oriented dialogue (MultiWOZ (Budzianowski et al., 2018)) and SQL query generation (Wik-iSQL (Zhong et al., 2017)) to form the task se-

quences for LSG on random tasks. For each scenario, we randomly select four different orders[2] (Appendix A.4) and run experiments for every order five times with different random seeds (20 runs per scenario). For each order, we report the average of all learned tasks' performance scores following Zhang et al. (2022); see Appendix A.5 for details of task-specific evaluation metrics.

### 5.2 Methods Compared

Following Zhang et al. (2022), we use GPT-2 (Radford et al., 2019) as the backbone model and Adapter (Houlsby et al., 2019) as the insertable module, and compare with the following methods:

- **Finetune** tunes the whole GPT-2 model only on the training data of the new task during the LSG process.

- **EWC** (Kirkpatrick et al., 2017) constrains the update of parameters that are important to previously learned tasks to alleviate forgetting.

- **LAMOL** (Sun et al., 2020) tunes the whole GPT-2 model with pseudo-sample replay.

- **Metac-Adapt (Metac)** (Wang et al., 2023) adapts LAMOL towards better semantic space for generating pseudo samples.

- **Adapter+LAMOL** only inserts adapter modules for the first task and tunes these modules with pseudo-sample replay while keeping the backbone model frozen.

- **AdapterCL** (Madotto et al., 2021) inserts task-

---

[2]Zhang et al. (2022) sample data from the original set for data balance. To ensure a fair comparison among all methods, we resample new data for experiments.

| Method | Similar Tasks | Random Tasks |
|---|---|---|
| DMEA | **65.8** | **57.3** |
| *w.o.* transfer | 64.9 | 56.5 |
| *w.o.* scaling | 65.5 | 56.8 |
| *w.o.* initialization | 65.4 | 57.0 |

Table 2: The average performance score for different ablations: (i) without forward knowledge transfer, (ii) without dynamic gradient scaling, and (iii) without dynamic initialization. All components improve the performance of our method.

| Time Step | | AdapterCL | ACM | DMEA |
|---|---|---|---|---|
| Similar | 2 | 55.8(+0.0) | 56.0(+0.1) | **56.3(+0.3)** |
| | 3 | 58.6(+0.0) | 59.1(+0.4) | **59.5(+0.6)** |
| | 4 | 61.2(+0.0) | 62.5(+0.6) | **63.2(+0.9)** |
| | 5 | 63.4(+0.0) | 64.8(+0.3) | **65.8(+1.0)** |
| Random | 2 | 55.4(+0.0) | 56.3(+0.9) | **57.1(+2.1)** |
| | 3 | 58.4(+0.0) | 58.9(+0.3) | **59.7(+1.3)** |
| | 4 | 64.0(+0.0) | 64.3(+0.7) | **65.4(+1.4)** |
| | 5 | 64.2(+0.0) | 64.9(+0.6) | **65.6(+1.1)** |

Table 3: The average performance score and forward knowledge transfer (FKT) of different methods at every time step. FKT is reported in parentheses.

specific adapter modules for every new task while keeping the backbone model and previous modules frozen.

- **ACM** (Zhang et al., 2022) dynamically adds adapter modules for new tasks depending on whether there are reusable previous modules to improve the performance and parameter efficiency of AdapterCL. It is the state-of-the-art on LSG.

# 6 Results and Analysis

## 6.1 Main Results

Table 1 shows the average performance score for each task sequence after learning all tasks (see Appendix A.7 for the performance of each task). From the results, we can see that DMEA outperforms previous baselines in all LSG settings, which demonstrates the superiority of our method. Note that while the learnable parameters of LAMOL are orders of magnitude larger, DMEA still achieves better performance than LAMOL in 7 out of 8 runs, showing its effectiveness in LSG.

Simply fine-tuning the model with new samples leads to poor performance due to catastrophic forgetting. Although EWC adopts Fisher information matrix to alleviate forgetting, its performance is still much worse than other memory-based baselines, indicating the importance of pseudo-sample replay. When learning from a sequence of similar tasks, Adapter+LAMOL performs better than AdapterCL as AdapterCL applies parameter isolation to different tasks which might prevent positive knowledge transfer across tasks. However, this is not the case when learning from random tasks: AdapterCL achieves much better results than Adapter+LAMOL as AdapterCL can avoid catastrophic forgetting by assigning different learnable parameters to each task. The performance of ACM is superior to Adapter+LAMOL and AdapterCL in both scenarios, showing the effectiveness of

its adaptive compositional architecture. However, ACM has no explicit mechanism to encourage forward knowledge transfer in LSG, which is actually the human learning paradigm. Our proposed DMEA consistently outperforms ACM by dynamically leveraging previously acquired knowledge to facilitate adaptation to new tasks.

## 6.2 Ablation Study

We conduct several ablations to analyze the contribution of different components of DMEA. In particular, we investigate three variants of DMEA (*a*) without selecting similar previous tasks for forward knowledge transfer (*w.o.* transfer), (*b*) removing dynamic gradient scaling (*w.o.* scaling), and (*c*) without dynamically initializing learnable coefficients (*w.o.* initialization). For each scenario, *i.e.,* similar tasks or random tasks, we randomly pick one sequence for experiments. Table 2 reports the average performance score after learning all tasks for different ablations.

From the results, we can observe that all components contribute to the average performance. Removing forward knowledge transfer leads to a significant performance drop in both scenarios, indicating that selecting top-$K$ most similar previous tasks can indeed discover and transfer useful learned knowledge to facilitate adaptation to the new task. The adoption of dynamic gradient scaling yields a moderate performance boost as it can balance the learning of the new task and replayed tasks to mitigate catastrophic forgetting. Dynamic initialization of learnable coefficients also facilitates performance improvement, demonstrating the effectiveness of leveraging the similarity of word frequency distributions between tasks.

## 6.3 Further Analysis

**Quantify Forward Knowledge Transfer.** Following Ke et al. (2020), we define metrics quanti-

| Metrics | Similar Tasks | Random Tasks |
|---|---|---|
| Input Subspace | **65.8** | **57.3** |
| Frequency | 65.3 | 56.9 |
| Representation | 65.2 | 56.9 |
| *w.o.* transfer | 64.9 | 56.5 |

Table 4: The average performance score using different similarity metrics.

fying forward knowledge transfer (FKT) at every time step $t$ as:

$$\text{FWT} = \frac{1}{t-1} \sum_{i=2}^{t} R_{i,i} - \bar{d}_i. \qquad (10)$$

where $R_{i,j}$ is the performance score on $\mathcal{T}^j$ after learning $\mathcal{T}^i$ and $\bar{d}_i$ refers to the performance of training $\mathcal{T}^i$ individually, which is actually the result of AdapterCL. For each scenario, we randomly select one sequence for analysis and report the average performance score along with FKT at each step in Table 3. From the results, we can see that DMEA consistently outperforms ACM in terms of the average performance score and FKT at all steps, demonstrating that DMEA can better facilitate positive knowledge transfer.

**Input Subspace vs. Other Similarity Metrics.** The ablation (*w.o.* transfer) in §6.2 demonstrates the importance of selecting similar learned tasks. To further investigate whether different similarity metrics influence the performance of DMEA, we conduct controlled experiments with two new metrics: (*a*) cosine similarity of word frequency distributions between different tasks (*frequency*), and (*b*) cosine similarity of the representations of selected samples from different tasks[3] (*representation*). For each scenario, we use the same sequence as §6.2. From the results in Table 4, we can observe that selecting similar previous tasks by input subspace consistently outperforms using other similarity metrics, demonstrating its superiority.

**Robustness to Module Type** To verify whether the performance gain of DMEA is consistent across different types of modules, we extend the experiments to prefix-tuning (Li and Liang, 2021) and LoRA (Hu et al., 2022). We randomly pick four sequences for experiments and report the average result in Table 5. we can see that DMEA still outperforms ACM when using other architecture as

---

[3]For a pair of tasks, we compute the cosine similarity for every representation pair and use the average as the similarity.

| Module Type | ACM | DMEA |
|---|---|---|
| Prefix-tuning | 62.6 | **63.4** |
| LoRA | 63.1 | **64.2** |

Table 5: The average performance score of ACM and DMEA with different module types.

the insertable module, showing its robustness to module type.

**Longer Sequence.** As mentioned in §5.1, we mainly conduct experiments on sequences consisting of 5 tasks following Zhang et al. (2022). To verify whether DMEA can still outperform the baselines when learning from a larger number of tasks, we further combine all tasks investigated in this work to form a longer sequence of 8 tasks. We evaluate ACM and DMEA on this longer sequence with 3 different orders and report the average performance score for each order after learning all tasks in Fig. 3. We can observe that DMEA is still superior to ACM when learning from longer sequences.

**Quality of Pseudo Data** Fig. 4 shows several pseudo samples generated by DMEA. We can see that DMEA can indeed generate high-quality pseudo samples to mitigate the forgetting of previously learned knowledge. However, the generated pseudo data could also be noisy as shown at the bottom of the figure, which might hinder further performance improvement.

**Other Types of Tasks** To explore whether the performance gain of DMEA is consistent on other types of tasks, we further include three new tasks: sentiment analysis (SST (Socher et al., 2013)), semantic role labeling (SRL (He et al., 2015)) and question answering (SQuAD (Rajpurkar et al., 2016)). We randomly select two tasks from the original task set three times and combine them with new tasks to form three task sequences. From the results shown in Table 6, we can observe that DMEA performs better than ACM on all sequences, showing its robustness to task types.

**Different Pseudo-data Sampling Ratios** Following Zhang et al. (2022), we set the pseudo-data sampling ratio to 0.2. To validate whether different pseudo-data sampling rates influence the performance gain of DMEA, we conduct controlled experiments with sampling rates $\{0.05, 0.1, 0.4\}$. We randomly pick three sequences for experiments and report the performance comparison between

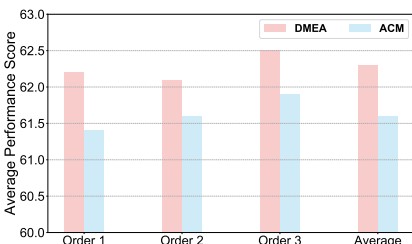

Figure 3: The average performance score for every order after learning all 8 tasks of the longer sequence.

---

**High-quality Data**

name[The Cricketers], eatType[restaurant], food[French], priceRange[moderate], near[Rainbow Vegetarian Cafe] what is the natural language form? The Cricketers is a French restaurant next to the Rainbow Vegetarian Cafe with moderate prices and a French taste.

---

**Noisy Data**

the table has columns week number, date, opponent, result, record and key words max, min, count, sum, avg, =, >, <, op, select, where, and, col, table, caption - - who the opponent was on the weekend where the record was 0 − 0? what is the translation from english to sql? select opponent from table where record = 0

---

Figure 4: Some examples of generated pseudo data. We color the task instruction in blue and output text in gray. Missing/wrong information is colored in red.

ACM and DMEA in Table 7. We can see that DMEA consistently outperforms ACM in all cases, demonstrating its effectiveness.

In addition, we show case studies of learned model architecture, model output, dynamic gradient scaling and task selection, generalization of dynamic initialization, and potential real-world applications in Appendix A.9 ∼ A.14, respectively.

## 7 Conclusion

In this work, we have introduced DMEA for lifelong sequence generation (LSG). DMEA leverages task correlations to dynamically determine the suitable architecture required to acquire novel knowledge of a new task and selects the most similar previous tasks through input subspace to facilitate knowledge transfer. It uses pseudo-sample replay along with dynamic gradient scaling to balance the learning of the new task and replayed tasks to further alleviate forgetting. With extensive experiments and analysis we have shown that DMEA consistently outperforms previous methods in different LSG settings. In the future, we would like to investigate ways to improve the quality of pseudo data and explore more metrics for task similarity.

| Method | Sequence | | | Average |
|--------|------|------|-------|---------|
| | (i) | (ii) | (iii) | |
| ACM | 68.4 | 63.6 | 71.8 | 67.9 |
| DMEA | **69.5** | **64.4** | **73.0** | **69.0** |

Table 6: The average performance score for every sequence after learning all new types of tasks.

| Sampling Ratio | 0.05 | 0.1 | 0.4 |
|--------|------|------|------|
| ACM | 61.7 | 62.0 | 62.1 |
| DMEA | **62.5** | **63.1** | **62.8** |

Table 7: The average performance score of ACM and DMEA with different pseudo-data sampling ratios.

## Limitations

Although effective, DMEA has couple of limitations:

- DMEA mainly focuses on the setting where every task has plenty of training samples. In contrast, humans can easily learn to perform new tasks with only few data, which is a hallmark of human intelligence. We leave how to explore lifelong sequence generation in few-shot settings as future work.

- DMEA does not consider machine translation, a sequence generation task that might involve vocabulary changes. One potential solution is to use multilingual pre-trained language models.

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

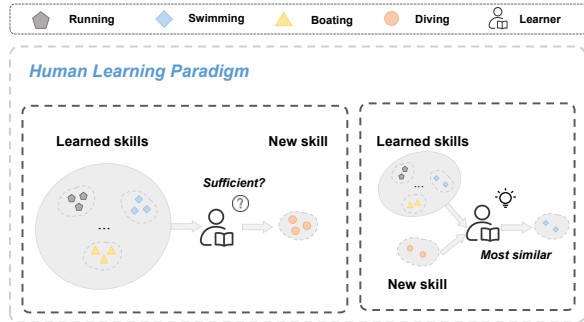

Figure 5: Given three learned skills, *i.e.,* swimming, running and boating, humans can determine that these skills are *not* sufficient for diving. And after realizing that swimming is the most similar learned skill, they only need to learn the new aspect, *i.e.,* how to safely jump off the diving platform to master the new diving skill.

Thomas Wolf, Lysandre Debut, Victor Sanh, Julien Chaumond, Clement Delangue, Anthony Moi, Pierric Cistac, Tim Rault, Remi Louf, Morgan Funtowicz, Joe Davison, Sam Shleifer, Patrick von Platen, Clara Ma, Yacine Jernite, Julien Plu, Canwen Xu, Teven Le Scao, Sylvain Gugger, Mariama Drame, Quentin Lhoest, and Alexander Rush. 2020. Transformers: State-of-the-art natural language processing. In *Proceedings of the 2020 Conference on Empirical Methods in Natural Language Processing: System Demonstrations*, pages 38–45, Online. Association for Computational Linguistics.

Friedemann Zenke, Ben Poole, and Surya Ganguli. 2017. Continual learning through synaptic intelligence. In *Proceedings of the 34th International Conference on Machine Learning, ICML 2017, Sydney, NSW, Australia, 6-11 August 2017*, volume 70 of *Proceedings of Machine Learning Research*, pages 3987–3995. PMLR.

Yanzhe Zhang, Xuezhi Wang, and Diyi Yang. 2022. Continual sequence generation with adaptive compositional modules. In *Proceedings of the 60th Annual Meeting of the Association for Computational Linguistics (Volume 1: Long Papers)*, pages 3653–3667, Dublin, Ireland. Association for Computational Linguistics.

Victor Zhong, Caiming Xiong, and Richard Socher. 2017. Seq2sql: Generating structured queries from natural language using reinforcement learning. *arXiv preprint arXiv:1709.00103*.

# A Appendix

## A.1 Illustration of Human Learning

We show the illustration of human learning in Fig. 5.

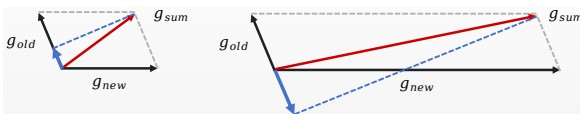

Figure 6: The effect of large gradient norm. $g_{old}$, $g_{new}$, and $g_{sum}$ represent the gradient of replayed tasks, the gradient of the new task, and the aggregated gradient, respectively. The blue arrows show the projection of $g_{sum}$ onto $g_{old}$. If the gradient norm of $g_{new}$ is large (right part of the figure), this projection might deviate too much from $g_{old}$.

> name[The Vaults], eatType[pub], priceRange[moderate], customer rating[1 out of 5], near[Caf Adriatic] what is the natural language form? A moderately priced pub, named The Vaults, is located near Caf Adriatic. It has a customer rating of 1 out of 5.

Figure 7: An example of the task instruction for E2ENLG. We color the task instruction in blue.

## A.2 Effect of Large Gradient Norm

As shown in the Fig. 6, if the gradient norm of the new task $g_{new}$ is large, the projection of the aggregated gradient $g_{sum}$ onto the gradient of replayed tasks $g_{old}$ might deviate too much from $g_{old}$, leading to more severe forgetting.

## A.3 Task Instruction Example

Following Zhang et al. (2022), we insert a natural language question describing the purpose of every task (task instruction) after the input of each sample. Fig. 7 shows an example of the task instruction for E2ENLG (Novikova et al., 2017).

## A.4 Task Orders

We present different task orders for two LSG scenarios in Table 8.

## A.5 Task-specific Evaluation Metrics

We report details of task-specific evaluation metrics in Table 9.

## A.6 Implementation Details

All methods are implemented with PyTorch/Transformers library (Wolf et al., 2020). We adopt AdapterHub (Pfeiffer et al., 2020) to implement adapter modules. For hyperparameters, we mainly follow the settings in Zhang et al. (2022) to have a fair comparison. In the expansion stage, we train the model for 6 epochs before selecting modules. In the adaptation stage, we set the number ($n$) of samples selected to obtain the

| Order | | Task Sequence |
|---|---|---|
| Similar Tasks | # 1 | e2e → res → hotel → tv → laptop |
| | # 2 | e2e → tv → res → laptop → hotel |
| | # 3 | res → hotel → e2e → laptop → tv |
| | # 4 | laptop → hotel → res → tv → e2e |
| Random Tasks | # 1 | mwoz → cnn → e2e → res → hotel |
| | # 2 | e2e → sql → hotel → mwoz → res |
| | # 3 | cnn → hotel → sql → e2e → mwoz |
| | # 4 | e2e → mwoz → laptop → sql → tv |

Table 8: Different task orders for each scenario. 'e2e' stands for E2ENLG. 'res', 'hotel', 'laptop' and 'tv' are four domains in RNNLG (restaurant, hotel, laptop and television). 'sql', 'cnn' and 'mwoz' respectively stand for 'WikiSQL' (SQL query generation), 'CN-NDM' (summarization) and 'MultiWOZ' (task-oriented dialogue).

| Dataset | Metric |
|---|---|
| RNNLG E2ENLG CNNDM | A-RG |
| WikiSQL | lfEM |
| MultiWOZ | dsEM |

Table 9: Details of task-specific evaluation metrics. 'A-RG', 'lfEM' and 'dsEM' respectively stand for 'average of ROUGE-1, ROUGE-2 and ROUGE-L scores', 'exact match of logical forms' and 'exact match of dialogue state'.

input subspace as 100. The threshold $\epsilon$ is set as 0.95 for selecting left-singular vectors. We adopt 1 for the number of similar tasks $K$. For dynamic gradient scaling, we set 100 for the number ($q$) of examples selected to calculate the gradient.

## A.7 Performance of Each Task

Table 10 shows the performance of each task for every task sequence after learning all tasks.

## A.8 Number of Learnable Parameters and Computational Resources

We present the average number of learnable parameters and average running time for ACM and DMEA in Table 11. From the comparison, we can observe that DMEA can outperform ACM with a negligible increase in learnable parameters and computational resources.

## A.9 Learned Model Architecture

To further demonstrate that dynamically initializing learnable coefficients can facilitate finding the

| Similar #1 | e2e | res | hotel | tv | laptop | Avg |
|---|---|---|---|---|---|---|
| ACM | 49.6 | 65.7 | 65.8 | 71.6 | 71.1 | 64.8 |
| DMEA | 49.2 | 67.1 | 68.1 | 72.5 | 72.0 | 65.8 |
| **Similar #2** | e2e | tv | res | laptop | hotel | **Avg** |
| ACM | 48.7 | 74.0 | 64.4 | 72.6 | 65.0 | 64.9 |
| DMEA | 47.9 | 74.9 | 64.9 | 74.2 | 65.9 | 65.6 |
| **Similar #3** | res | hotel | e2e | laptop | tv | **Avg** |
| ACM | 65.6 | 67.3 | 48.5 | 72.0 | 69.3 | 64.5 |
| DMEA | 66.9 | 66.6 | 49.5 | 73.7 | 70.9 | 65.5 |
| **Similar #4** | laptop | hotel | res | tv | e2e | **Avg** |
| ACM | 73.1 | 66.8 | 66.9 | 72.4 | 47.6 | 65.4 |
| DMEA | 74.6 | 67.6 | 67.4 | 72.9 | 48.3 | 66.2 |
| **Random #1** | mwoz | cnn | e2e | res | hotel | **Avg** |
| ACM | 81.6 | 26.0 | 47.5 | 64.5 | 64.1 | 56.7 |
| DMEA | 81.6 | 26.5 | 48.1 | 65.7 | 65.4 | 57.5 |
| **Random #2** | e2e | sql | hotel | mwoz | res | **Avg** |
| ACM | 48.4 | 62.7 | 64.6 | 84.8 | 64.0 | 64.9 |
| DMEA | 48.7 | 64.9 | 64.9 | 84.9 | 64.6 | 65.6 |
| **Random #3** | cnn | hotel | sql | e2e | mwoz | **Avg** |
| ACM | 26.3 | 63.2 | 62.1 | 47.7 | 82.4 | 56.3 |
| DMEA | 26.6 | 65.0 | 63.1 | 48.4 | 83.5 | 57.3 |
| **Random #4** | e2e | mwoz | laptop | sql | tv | **Avg** |
| ACM | 48.6 | 80.5 | 70.3 | 63.5 | 68.2 | 66.2 |
| DMEA | 49.0 | 82.8 | 71.7 | 64.4 | 68.9 | 67.4 |

Table 10: The performance of each task for every sequence after learning all tasks.

| Method | Avg Para Num | Avg Time (min) |
|---|---|---|
| ACM | 4.6M | 218.1 |
| DMEA | 4.8M | 223.5 |

Table 11: The comparison of the average number of learnable parameters (Avg Para Num) and average running time (Avg Time) between ACM and DMEA.

optimal model architecture, we analyze the model expansion stage of ACM and DMEA using sequence #4 in *random* scenario. For the final task *tv*, ACM decides to reuse modules from the first (*e2e*) and the third task (*laptop*) while DMEA reuses all modules from *laptop* which is consistent with the observation that the similarity between *tv* and *laptop* is much higher than that between *tv* and *e2e*.

## A.10 Case Study of Model Output

We select RNNLG.hotel (sequence #1 in *similar* scenario) and WikiSQL (sequence #4 in *random* scenario) as two representative tasks and show several examples of output in Table 12. Compared with ACM, DMEA possesses the capability to convey more precise and relevant information from the input without introducing superfluous details.

| **RNNLG.hotel:** inform(name='mandarin oriental san francisco';dogsallowed='yes';pricerange='ultra high end') | |
|---|---|
| Reference | *the mandarin oriental san francisco is in the ultra high end price range and allows dogs.* |
| ACM | *the mandarin oriental san francisco is a hotel in the ultra high end range (missing: and allows dogs).* |
| DMEA | *the mandarin oriental san francisco offers ultra high end accommodations and allows dogs.* |
| **WikiSQL:** on which date was the winning driver alain prost and had damon hill in the pole position ? | |
| Reference | *select date from table where winning driver = alain prost and pole position = damon hill* |
| ACM | *select date from table where winning driver = alain prost and pole position = damon (missing: hill)* |
| DMEA | *select date from table where pole position = damon hill and winning driver = alain prost* |

Table 12: Output examples of different methods after learning the whole sequence. We color missing/wrong information in red and redundant information in blue.

| Method | Similar Tasks | Random Tasks |
|---|---|---|
| ACM | 64.5 | 66.2 |
| ACM *w* DI | **64.7** | **66.5** |

Table 13: The performance comparison between ACM and ACM with dynamic initialization (ACM *w* DI).

## A.11 Case Study of Dynamic Gradient Scaling

The ablation study in §6.2 demonstrates the importance of dynamic gradient scaling. We further conduct a case study using sequence #1 in *random* scenario. During the learning of this sequence, the fourth task *res* reuses several modules from the third task *e2e*. After applying dynamic gradient scaling, the performance of *e2e* is improved by 0.3 without compromising *res*, indicating that it does mitigate the bias towards the new task.

## A.12 Case Study of Task Selection

To verify that the previous task chosen in the selection stage is indeed the most similar to the new task, we analyze several cases using sequence #2 in *random* scenario. For the third task *hotel*, the selected first task *e2e* has the highest similarity score as they share the same task type. In addition, the third task *hotel* shares a similar semantic space with the final task *res*. Therefore, it is selected for forward knowledge transfer when learning *res*.

## A.13 Generalization of Dynamic Initialization

To demonstrate the generalization ability of dynamic initialization, we apply it to the expansion stage of ACM. For each scenario, we randomly pick one sequence for experiments. As reported in Table 13, dynamic initialization does benefit ACM, verifying its generalization capability.

## A.14 Real World Application

Apart from the aforementioned sequence generation tasks, DMEA demonstrates the potential to be applied to various real-world lifelong learning scenarios. For example, it can continually train a model to perform summarization and question-answering based on news articles from different domains during the onset of an emerging event like Covid-19.

## A.15 Hyperparameter Search

We select the number of training epochs before modules selection from $\{6, 9, 12\}$, the number ($n$) of samples picked to obtain the input subspace from $\{50, 100, 200, 500\}$ and the threshold $\epsilon$ for selecting left-singular vectors from $\{0.90, 0.95, 0.99\}$. The number of similar previous tasks $K$ is selected from $\{1, 2, 3\}$. The number ($q$) of examples for calculating the gradient in dynamic gradient scaling is selected from $\{20, 50, 100, 200\}$.