# OpenReview forum: "Lifelong Sequence Generation with Dynamic Module Expansion and Adaptation"
_EMNLP/2023/Conference — EMNLP 2023 Main_

### Official Review · Reviewer_kDLs · 2023-08-03

**Soundness:** 4

**Excitement:**

3: Ambivalent: It has merits (e.g., it reports state-of-the-art results, the idea is nice), but there are key weaknesses (e.g., it describes incremental work), and it can significantly benefit from another round of revision. However, I won't object to accepting it if my co-reviewers champion it.

**Paper Topic And Main Contributions:**

This paper focuses on lifelong learning for sequence generation. The authors introduce a novel framework that can dynamically determine whether to add new modules for acquiring novel knowledge from new tasks. Additionally, they suggest retrieving knowledge from prior tasks to enhance adaptation. The authors have carried out comprehensive experiments in two lifelong learning settings: (1) identical tasks across different domains, and (2) distinct, randomly ordered tasks. In most instances, the proposed methods outperform the baselines. Overall, I find the proposed method reasonable, and the experimental results support it well. However, the authors claim this is the first work to address LSG from a human learning standpoint. To my knowledge, most lifelong learning studies purport to be inspired by human learning. This claim might be overstated, but I don't believe it diminishes the contributions made by the authors.

**Questions For The Authors:**

1. What is the model performance with different $K$ and number of samples picked to obtain the input subspace $n$?

2. As TRGP is already a powerful continual learning method, why did you not include it as a baseline? Also, I would suggest you to compare with some other continual learning baselines, such as gradient projection memory [1].

I would like to increase my score if the questions are well answered.

[1] Saha, Gobinda, Isha Garg, and Kaushik Roy. "Gradient projection memory for continual learning." arXiv preprint arXiv:2103.09762 (2021).

**Reasons To Accept:**

1. This paper is clearly written and easy to follow.

2. The idea of using knowledge of previous tasks is novel in lifelong sequence generation tasks.

3. The experiments are comprehensive and support the claim.


**Reasons To Reject:**

1. The claim that this is the first work to address LSG from a human learning perspective is aggressive. I suggest highlighting more on the methodology part.

2. More hyper-parameter analysis is needed. The authors mention the search space of hyper-parameters in Appendix A.18, but I would like to see more analysis about how the hyper-parameters affect the performance, for example, K.


**Reproducibility:**

4: Could mostly reproduce the results, but there may be some variation because of sample variance or minor variations in their interpretation of the protocol or method.

**Reviewer Confidence:**

4: Quite sure. I tried to check the important points carefully. It's unlikely, though conceivable, that I missed something that should affect my ratings.

---

> ### Author Rebuttal · Authors · 2023-08-28
>
> Dear Reviewer:
>
> Thank you for the positive assessment and insightful comments.
>
> **Q1. The claim that this is the first work to address LSG from a human learning perspective is aggressive. I suggest highlighting more on the methodology part.**
>
> Thanks for your suggestion! We will highlight more on the methodology part in the final version.
>
>
> **Q2. More hyper-parameter analysis is needed. The authors mention the search space of hyper-parameters in Appendix A.18, but I would like to see more analysis about how the hyper-parameters affect the performance, for example, K. / What is the model performance with different K and number of samples picked to obtain the input subspace n?**
>
> We select hyperparameters using a sequence that is not used for final evaluation (Table 1). The model performance with different K and n is shown below. For the number of similar tasks K, the model achieves the best performance with K=1, which should be because selecting too many previous tasks introduces unnecessary noise. For the number (n) of samples selected to obtain the input subspace, increasing n does not always improve the performance. When n reaches 100, the performance tends to be stable. Therefore, we use n=100 to reduce computational cost.
>
> | K  | 1 | 2 | 3 |
> |  ----  | ----  | ----  | ----  |
> | Performance | 59.4  | 59.0 | 58.8 |
>
> | n  | 50 | 100 | 200 | 500 |
> |  ----  | ----  | ----  | ----  | ----  |
> | Performance | 58.9  | 59.4 | 59.4 | 59.4 |
>
> **Q3. As TRGP is already a powerful continual learning method, why did you not include it as a baseline? Also, I would suggest you to compare with some other continual learning baselines, such as gradient projection memory [1].**
>
> To better demonstrate the effectiveness of DMEA, we further compare DMEA with two methods originally designed for classification: GPM[1], and TRGP [2]. We randomly select four task sequences for pilot experiments. From the average results shown below, we can see that DMEA outperforms both methods by a large margin, demonstrating the effectiveness of pseudo-sample replay and forward knowledge transfer. We will conduct more extensive experiments in the final version.
>
> | DMEA  | GPM | TRGP |
> |  ----  | ----  | ----  |
> | 62.9 | 59.1  | 59.6 |
>
> Thanks again for your feedback. We hope that this addresses your questions.
>
> [1] Gradient projection memory for continual learning. Saha, Gobinda, Isha Garg, and Kaushik Roy. ICLR 2021
>
> [2] TRGP: Trust Region Gradient Projection for Continual Learning. Lin, Sen, et al. ICLR 2022

---

### Official Review · Reviewer_X4NL · 2023-08-03

**Soundness:** 4

**Excitement:**

4: Strong: This paper deepens the understanding of some phenomenon or lowers the barriers to an existing research direction.

**Paper Topic And Main Contributions:**

Lifelong sequence generation (LSG) is a research area in continual learning that aims to train a model on a sequence of generation tasks while avoiding forgetting previously learned knowledge and continuously acquiring new generation patterns. As mentioned earlier, the biggest problem with continual learning is forgetting formerly known knowledge. This article proposes Dynamic Module Expansion and Adaptation, which allows the model to dynamically determine its architecture for acquiring new knowledge based on the relationships between tasks and select the most similar previous study to adapt to the new job or add new modules. Finally, dynamic gradient scaling is applied to balance the weight of the current task and previously learned lessons, mitigating the forgetting problem.
I think the main contribution of this article is that it opens up a new approach to exploring the forgetting problem in continual learning. Through this approach, it is possible to mitigate the model's bloating that comes with continuously adding modules, i.e., keeping the model's parameters from becoming too large while maintaining good performance. Compared to the trend of increasing the size of models, incorporating more thinking can reduce parameters and deepen our understanding of the problem in this field.


**Reasons To Accept:**

This article provides a new approach to solving the forgetting problem in continual learning. Unlike previous works, this work takes a novel perspective from human thinking and can somewhat mitigate the forgetting problem. Therefore, it has a degree of innovation. Additionally, the article provides a detailed explanation of the formulas, making it easier for people to understand. Finally, this article also conducts complex experiments and comparisons with different models, showing some improvements in the experimental results.

**Reasons To Reject:**

From my perspective, the experimental section of the paper could benefit from conducting some ablation experiments. For example, if the paper proposes three stages to handle multi-task continual learning, it would be helpful to determine which location has the most significant impact on the final results and analyze the reasons for this. This would help us to understand the work in this area better.

**Reproducibility:**

4: Could mostly reproduce the results, but there may be some variation because of sample variance or minor variations in their interpretation of the protocol or method.

**Reviewer Confidence:**

3: Pretty sure, but there's a chance I missed something. Although I have a good feel for this area in general, I did not carefully check the paper's details, e.g., the math, experimental design, or novelty.

---

> ### Author Rebuttal · Authors · 2023-08-28
>
> Dear Reviewer:
>
> Thank you for the positive assessment and insightful comments.
>
> **Q1. From my perspective, the experimental section of the paper could benefit from conducting some ablation experiments. For example, if the paper proposes three stages to handle multi-task continual learning, it would be helpful to determine which location has the most significant impact on the final results and analyze the reasons for this. This would help us to understand the work in this area better.**
>
> We do provide several ablations to analyze our method. For example, we investigate three variants of our method in Section 6.2 (line 519-545):
>
> (1) without selecting similar previous tasks for forward knowledge transfer (the whole selection stage)
>
> (2) removing dynamic gradient scaling (in the adaptation stage)
>
> (3) without dynamically initializing learnable coefficients (in the expansion stage)
>
> According to the results in Table 2, forward knowledge transfer plays the most important role in the final performance, demonstrating the necessity of discovering and transferring useful learned knowledge to facilitate adaptation to the new task in lifelong sequence generation (the effectiveness of forward knowledge transfer is also verified in Table 3).
>
> In addition, we further analyze different aspects of our method in Section 6.3 and Appendix, e.g., similarity metrics, module types, and pseudo-data sampling ratios.
>
> Thanks again for your feedback. We hope that this addresses your questions.

---

### Official Review · Reviewer_9rh6 · 2023-08-04

**Typos Grammar Style And Presentation Improvements:** N/A
**Soundness:** 3

**Excitement:**

2: Mediocre: This paper makes marginal contributions (vs non-contemporaneous work), so I would rather not see it in the conference.

**Missing References:**

See above

**Paper Topic And Main Contributions:**

This paper concerns continual learning in NLP, in particular the knowledge transfer issues.

**Questions For The Authors:**

See above

**Reasons To Accept:**

The problem concerned is important and valid. I believe the NLP community should pay more attention on it.

**Reasons To Reject:**

1. When talking about task similarity in continual learning, a popular paper is CAT [1], which considers not only similar tasks and dissimilar tasks, but also the mixture of them. Have you ever tried this challenging scenario and any interesting observations?
2. I think there are more systems to be compared. While you mentioned some systems are focused on classification, I do not see why they cannot be a baseline in generation setting by simply changing the underlying LM and classification head (in particular for those baselines based on adapter). I think the author should at least compare those NLP methods that are explicitly dealing with transfer (as cited by the author in the related work, like [2] or even more in the survey [4]).
3. Similarity detection has been used in different systems [1,3], what are the main advantages for DMEA compared to them? Or is it just another heuristic method?

[1]: Continual Learning of A Mixed Sequence of Similar and Dissimilar Tasks, NeurIPS2020
[2]: Achieving Forgetting Prevention and Knowledge Transfer in Continual Learning, NeurIPS2021
[3]: Beyond not-forgetting: Continual learning with backward knowledge transfer, NeurIPS2022
[4]: Continual learning of natural language processing tasks: A survey https://arxiv.org/pdf/2211.12701.pdf

**Reproducibility:**

4: Could mostly reproduce the results, but there may be some variation because of sample variance or minor variations in their interpretation of the protocol or method.

**Reviewer Confidence:**

4: Quite sure. I tried to check the important points carefully. It's unlikely, though conceivable, that I missed something that should affect my ratings.

---

> ### Author Rebuttal · Authors · 2023-08-28
>
> Dear Reviewer:
>
> We thank you for your thoughtful comments and feedback. We address your questions here.
>
> **Q1. When talking about task similarity in continual learning, a popular paper is CAT [1], which considers not only similar tasks and dissimilar tasks, but also the mixture of them. Have you ever tried this challenging scenario and any interesting observations?**
>
> We do consider this challenging scenario:
>
> (1) As shown in Table 6, the ‘random’ scenario already includes both similar tasks and dissimilar tasks.
>
> (2) In Section 6.3, Longer Sequence (line 588-599), we further fuse all 8 tasks investigated in this work together for experiments. DMEA still outperforms ACM when learning from longer sequences.
>
>
> **Q2. I think there are more systems to be compared. While you mentioned some systems are focused on classification, I do not see why they cannot be a baseline in generation setting by simply changing the underlying LM and classification head (in particular for those baselines based on adapter). I think the author should at least compare those NLP methods that are explicitly dealing with transfer (as cited by the author in the related work, like [2] or even more in the survey [4]).**
>
> Thanks for your suggestions!
>
> Due to the time limit, we do not manage to change the code of [2] to GPT-2. We will try to include it in the final version.
>
> To better demonstrate the superiority of DMEA over other methods, we further compare DMEA with two methods: (1) an adapter-based method AdapterFusion [5], and (2) TRGP [6] which explicitly encourages forward knowledge transfer (originally designed for classification). We randomly select four task sequences for pilot experiments. The average performance score comparison between different methods is shown below. We can observe that DMEA achieves better performance than AdapterFusion and TRGP, demonstrating its effectiveness. We will conduct more extensive experiments in the final version.
>
> | DMEA  | AdapterFusion | TRGP |
> |  ----  | ----  | ----  |
> | 62.9 | 61.1  | 59.6 |
>
> **Q3. Similarity detection has been used in different systems [1,3], what are the main advantages for DMEA compared to them? Or is it just another heuristic method?**
>
> Compared to [1], the main advantage of DMEA is that [1] requires training another transfer model and reference model while DMEA only needs to calculate the subspace using the original model (more computation-efficient).
>
> Compared to [3], the main advantages of DMEA are (1) we directly use subspace projection instead of its proxy (gradient projection), which could better reflect the correlation between tasks. (2) [3] calculated gradient projection before learning the new task while DMEA trains the model on the new task for several epochs in the first expansion stage, which provides more task-specific knowledge for subsequent subspace calculation (Section 4.2).
>
> To support our claim, we replace the similarity detection method of DMEA with the method in [3] as it is relatively easy to implement (denoted as DMEA-G). We conduct pilot experiments using the same four task sequences as in **Q2**.  The average results are reported below, showing that our designed similarity detection method is superior to that in [3]. We will conduct more extensive experiments in the final version.
>
> | DMEA  | DMEA-G |
> |  ----  | ----  |
> | 62.9 | 62.5  |
>
> Thanks again for your feedback. We hope that this addresses your questions.
>
> [1]: Continual Learning of A Mixed Sequence of Similar and Dissimilar Tasks, NeurIPS2020
>
> [2]: Achieving Forgetting Prevention and Knowledge Transfer in Continual Learning, NeurIPS2021
>
> [3]: Beyond not-forgetting: Continual learning with backward knowledge transfer, NeurIPS2022
>
> [4]: Continual learning of natural language processing tasks: A survey, arXiv
>
> [5] AdapterFusion: Non-destructive task composition for transfer learning. Pfeiffer, Jonas, et al. EACL 2021
>
> [6] TRGP: Trust Region Gradient Projection for Continual Learning. Lin, Sen, et al. ICLR 2022

---

### Official Review · Reviewer_5YgP · 2023-08-05

**Typos Grammar Style And Presentation Improvements:** None.
**Soundness:** 4

**Excitement:**

4: Strong: This paper deepens the understanding of some phenomenon or lowers the barriers to an existing research direction.

**Missing References:**

Please refer to my second point in "Reason To Reject".

**Paper Topic And Main Contributions:**

This paper focuses on lifelong sequence generation (LSG) which aims to continually train a model on a sequence of generation tasks to learn constantly emerging new generation patterns. The authors extend the previous SoTA method ACM to propose Dynamic Module Expansion and Adaptation (DMEA) for LSG. DMEA follows the learning paradigm of humans by dividing the learning process of a new task into expansion, selection and adaptation. Such a design eases forgetting while enables knowledge transfer. Experimental results demonstrate the effectiveness of DMEA.

**Questions For The Authors:**

1. The proposed method doesn't need to fine-tune the base model. Although the experiments are done on GPT-2, I'm wondering whether it's possible to apply DMEA to LLM, e.g., GPT-3.

**Update**: After the rebuttal period, I raised my excitement score to 4.

**Reasons To Accept:**

1. The proposed method is well motivated, i.e., having a link with human learning paradigm. DMEA divides the learning process of a new task into expansion, selection and adaptation which not only eases the forgetting problem but also enables knowledge transfer.
2. DMEA gives new SoTA results by exceeding the performance of ACM. The experiment and analysis in the work are extensive.

**Reasons To Reject:**

1. The paper is a good introduction of new method but further insights on LSG or continual learning with PLMs are lacking. The proposed method is composed of several tricks, i.e., forward knowledge transfer module, dynamic gradient scaling, dynamic initialization, and the ablation study shows that each of them can contribute a small improvement to the system. However, according to Table 1, there still exists a gap between DMEA and MTL. It's unclear whether DMEA can be further extended to get close to MTL performance since DMEA is a nicely crafted system instead of a principle idea.
2. The related literature is not treated seriously. Although the "Related Work" section surveys Lifelong Learning and Lifelong Sequence Generation, I think more recent works which study lifelong learning / continual learning with PLM should be considered. For example, [1] studies the influence of generation objective on PLM in the continual learning process; [2] studies continual instruction tuning which also involves generation. Specifically, since the input data used in this work also includes the task description, it's similar to the continual instruction tuning setup and it's recommended to compare DMEA with method in [2] if possible.

[1] Class-Incremental Learning based on Label Generation, Shao et al., ACL 2023.

[2] ConTinTin: Continual Learning from Task Instructions, Yin et al., ACL 2022.

**Reproducibility:**

3: Could reproduce the results with some difficulty. The settings of parameters are underspecified or subjectively determined; the training/evaluation data are not widely available.

**Reviewer Confidence:**

5: Positive that my evaluation is correct. I read the paper very carefully and I am very familiar with related work.

---

> ### Author Rebuttal · Authors · 2023-08-28
>
> Dear Reviewer:
>
> Thank you for the positive assessment and insightful comments.
>
> **Q1. The paper is a good introduction of new method but further insights on LSG or continual learning with PLMs are lacking. The proposed method is composed of several tricks, i.e., forward knowledge transfer module, dynamic gradient scaling, dynamic initialization, and the ablation study shows that each of them can contribute a small improvement to the system. However, according to Table 1, there still exists a gap between DMEA and MTL. It's unclear whether DMEA can be further extended to get close to MTL performance since DMEA is a nicely crafted system instead of a principle idea.**
>
> Apart from the main results in Section 6.1, we do provide many insights on LSG in Section 6.3 and the Appendix. For example, we analyze the influence of different similarity metrics for selecting similar learned tasks on LSG and the robustness of DMEA to module types (can also indicate which module is more suitable for LSG) in Section 6.3. Besides, we show how different pseudo-data sampling rates affect LSG performance in Appendix A.10.
>
> Since MTL serves as the upper bound for LSG, it’s normal for DMEA to perform worse than MTL. Actually, DMEA is easy to extend. For example, it can be combined with a more advanced pseudo-sample generation method (AQF-RQ) [3]. We randomly select four task sequences for pilot experiments. From the average results shown below, we can observe that this simple extension can indeed improve the performance. We will conduct more extensive experiments in the final version.
>
> | DMEA  | DMEA + AQF-RQ |
> |  ----  | ----  |
> | 62.9 | 63.3  |
>
> In addition, although DMEA is a nicely crafted system, its component (dynamic initialization) can be easily applied to the baseline ACM to improve the performance as shown in Appendix A.16.
>
> **Q2. The related literature is not treated seriously. Although the "Related Work" section surveys Lifelong Learning and Lifelong Sequence Generation, I think more recent works which study lifelong learning / continual learning with PLM should be considered. For example, [1] studies the influence of generation objective on PLM in the continual learning process; [2] studies continual instruction tuning which also involves generation. Specifically, since the input data used in this work also includes the task description, it's similar to the continual instruction tuning setup and it's recommended to compare DMEA with method in [2] if possible.**
>
> Thanks for your suggestions!
>
> As [1] was submitted to Arxiv on 22 June 2023, we did not know about it during the EMNLP submission. We will cite it in the final version.
>
> For ConTinTin [2], we will cite it as related work in the final version. However, it’s difficult for us to compare DMEA with ConTinTin as (1) there is no open-source code for ConTinTin, and (2) ConTinTin requires instructions to contain specific content, e.g., “Things to avoid” and “negative examples”, which is not satisfied in our case.
>
> **Q3. The proposed method doesn't need to fine-tune the base model. Although the experiments are done on GPT-2, I'm wondering whether it's possible to apply DMEA to LLM, e.g., GPT-3.**
>
> We cannot apply DMEA to GPT-3 as it’s not possible (at the moment) to add learnable modules to GPT-3. However, we can apply DMEA to open-source LLMs. We choose GPT-J-6B as an example because its corresponding code is relatively easy to implement. We randomly select four task sequences for pilot experiments. The comparison of average results between DMEA and ACM is shown below. We can see that DMEA still outperforms ACM when using a larger language model as the backbone. We will conduct more extensive experiments in the final version.
>
> | DMEA  | ACM |
> |  ----  | ----  |
> | 66.3 | 64.9  |
>
> Thanks again for your feedback. We hope that this addresses your questions.
>
>
> [1] Class-Incremental Learning based on Label Generation, Shao et al., ACL 2023.
>
> [2] ConTinTin: Continual Learning from Task Instructions, Yin et al., ACL 2022.
>
> [3] Ask Question First for Enhancing Lifelong Language Learning. Wang H, Fu R, Zhang X, et al. COLING 2022

---

### Meta-Review · Area_Chair_jxsY · 2023-09-19

**Recommendation:** 5

**Metareview:**

The reviewers mostly agree that the paper is well-written and easy to follow. Also, most reviewers reach a consensus that the proposed method is novel and acknowledge that the method achieves new SoTA results. The reviewers also asked more analysis on the hyper-parameters, additional baselines, ablation studies on each parts and missing references. The authors mostly addressed the concerns raised by the reviewers except for the reviewer 9rh6. Reviewer 9rh6 could not provide further feedback or comments on the authors' rebuttal. Given the discussion, we weight more on the other reviewers' scores.

As a result, the reviewers mostly agree that the paper is 'strong' on the soundness. ([4,4,4], *3 - 9rh6*). In terms of excitement, two reviewers give 'Strong' rating, one 'Ambivalent' and one 'Mediocre'. Therefore, the excitement score is more towards 'strong'.

---

### Decision · Program_Chairs · 2023-10-07

**Decision:**

Accept-Main

**Comment:**

The reviewers mostly agree that the paper is well-written and easy to follow. Also, most reviewers reach a consensus that the proposed method is novel and acknowledge that the method achieves new SoTA results. The reviewers also asked more analysis on the hyper-parameters, additional baselines, ablation studies on each parts and missing references. The authors mostly addressed the concerns raised by the reviewers except for the reviewer 9rh6. Reviewer 9rh6 could not provide further feedback or comments on the authors' rebuttal. Given the discussion, we weight more on the other reviewers' scores.

As a result, the reviewers mostly agree that the paper is 'strong' on the soundness. ([4,4,4], *3 - 9rh6*). In terms of excitement, two reviewers give 'Strong' rating, one 'Ambivalent' and one 'Mediocre'. Therefore, the excitement score is more towards 'strong'.